# QUESTION-AWARE KNOWLEDGE GRAPH PROMPTING FOR LARGE LANGUAGE MODELS

## ABSTRACT

Large Language Models (LLMs) have demonstrated significant advancements in various natural language processing tasks, yet they often struggle with tasks that require external domain-specific knowledge, such as Multiple Choice Question Answering (MCQA). Integrating Knowledge Graphs (KGs) with LLMs has been explored as a solution to enhance LLMs' reasoning capabilities, while existing methods either involve computationally expensive finetuning processes or rely on the noisy retrieval of KG information. Recent efforts have focused on leveraging Graph Neural Networks (GNNs) to generate KG-based soft prompts for LLMs, which face challenges of lacking question-relevance assessment in GNN and utilization of relations among options. In this paper, we propose a novel approach, QAP, to address these challenges by optimizing the utilization of KG in MCQA tasks. Our method introduces question embeddings into the GNN aggregation process, enabling the model to assess the relevance of KG information based on the question context. Additionally, QAP facilitates inter-option interactions by employing an attention module that explicitly models relationships between answer options. Specifically, we use multiple attention heads for the GNN output, allowing the model to capture and compare features across different options, thereby enhancing cross-option reasoning. Our approach not only enhances the connection between GNNs and LLMs but also enables the model to better utilize the relationships between answer options. Experimental results demonstrate that QAP outperforms state-of-the-art models on multiple public MCQA datasets, validating its effectiveness and scalability.

## 1 INTRODUCTION

In recent years, pretrained Large Language Models (LLMs) (Brown et al., 2020; Touvron et al., 2023) have made significant strides in natural language processing (NLP) tasks (Wei et al., 2022b; Cohen et al., 2024; Chen et al., 2024). Leveraging vast amounts of data and computational resources, LLMs have demonstrated remarkable performance in tasks such as language generation (Cheng et al., 2023) and text comprehension (Lewis et al., 2020). However, despite their impressive achievements, LLMs still face challenges when it comes to tasks that require domain-specific knowledge or external information (Zheng et al., 2023; Wang et al., 2023). A typical example is the Multiple Choice Question Answering (MCQA) task, where the correct answer may rely on complex background knowledge that goes beyond what LLMs have learned from pretraining corpora (Asai et al., 2024). To address this limitation, researchers have started exploring ways to integrate external knowledge bases, such as Knowledge Graphs (KGs), into LLMs to enhance their reasoning and answering capabilities (Jiang et al., 2024; Sun et al., 2024).

Several existing studies have proposed to leverage KGs for assisting LLMs in answering questions (Jiang et al., 2023b; Ma et al., 2024). Some approaches incorporate KG information directly into the training or finetuning process of LLMs (Zhang et al., 2019; Wang et al., 2021). For instance, K-Adapter (Wang et al., 2021) introduces entity and relation knowledge during model training, leading to improved performance in knowledge reasoning tasks. However, such methods require retraining or finetuning the LLMs, which is computationally expensive and challenging to scale in resource-limited environments. Another class of methods retrieves relevant information from KGs and appends it to the LLM input, enabling the model to utilize this information during inference (Baek et al., 2023; Luo et al., 2024). While these approaches avoid finetuning, the retrieval quality is often suboptimal, especially when the retrieved KG content is not semantically

aligned with the question. This misalignment introduces noise and degrades the quality of generated answers (Xu et al., 2024). More recently, researchers have proposed to combine the benefits of finetuning and retrieving by utilizing KG-based soft prompts (Lester et al., 2021; Qin et al., 2021). KG-based soft prompts are lightweight and flexible input prefixes obtained from KGs using Graph Neural Networks (GNNs), which can guide LLM's output. However, existing KG-based soft prompt methods face two major limitations. First, GNNs lack incorporation of the target question, making it difficult for GNNs to assess the relevance of KG information to the question, leading to suboptimal information utilization. Second, existing methods for MCQA generate soft prompts for each answer option independently, without considering the relationships between different options. In practice, such relationships can help jointly evaluate all options and exclude incorrect options, ultimately reaching the correct answer.

To overcome these challenges, we propose a novel method, QAP (**Q**uestion-**A**ware Knowledge Graph **P**rompting), which generates KG-based soft prompts for LLM reasoning on MCQA tasks in a Question-Aware manner. Our approach addresses the first limitation by incorporating question embeddings into the GNN aggregation process, enabling the model to better assess the relevance of KG information to the question context. This improves the the model's utilization of the KG information and creates a stronger connection between the GNN and the question text. For the second limitation, we propose an Inter-Option Attention mechanism that allows for interactions among different answer options by mapping the GNN node representations to multiple option sequences, encouraging the model to leverage the relationships between these options. This approach is particularly beneficial in cases where individual option evaluation is difficult, while in contrast, considering all options together provides a clearer decision boundary. We present the comparison of different methods in Figure 1.

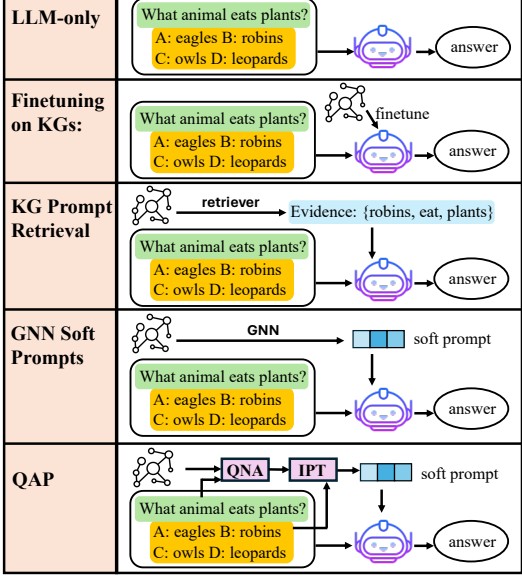

Figure 1: The landscape of existing methods for MCQA and our proposed method QAP. QAP utilizes our designed Question-Aware Neighborhood Aggregation (QNA) and Inter-Option Attention (IPT) modules to preform MCQA.

The contributions of our work can be summarized as follows:

- We study the challenges of KG-based soft prompt methods for MCQA associated with the lack of question-relevance assessment in GNN and the omission of relations among options.

- We propose QAP, a novel method for addressing the challenges of utilizing KG information in MCQA tasks. Our approach provides the question-relevance assessment in a Question-Aware Neighborhood Aggregation module (QNA) and uses an Inter-Option Attention module (IPT) to generate soft prompts, effectively leveraging the information from questions and options to improve the overall reasoning in MCQA.

- Experimental results show that QAP surpasses current state-of-the-art models across multiple public MCQA datasets, confirming its effectiveness and demonstrating its superiority in tackling domain-specific reasoning tasks.

## 2 PROBLEM FORMULATION

In this section, we introduce the task of Multiple Choice Question Answering (MCQA) based on knowledge graphs. We aim to answer a question $q$ by selecting one of the $n$ answer options from the candidate set $\mathcal{A} = \{a_k | k = 1, 2, \ldots, n\}$. This is performed with the assistance of a knowledge graph $\mathcal{G} = (\mathcal{E}, \mathcal{R}, \mathcal{T})$, where $\mathcal{E}$ and $\mathcal{R}$ are sets of entities and relations, respectively, and $\mathcal{T} = \{(h, r, t) | h, t \in \mathcal{E}, r \in \mathcal{R}\}$ is the set of knowledge triplets, each containing a head entity $h$, a relation $r$ and a tail entity $t$. We utilize a pretrained large language model denoted as $LM$ to generate the final answer to the question $q$, which is the input.

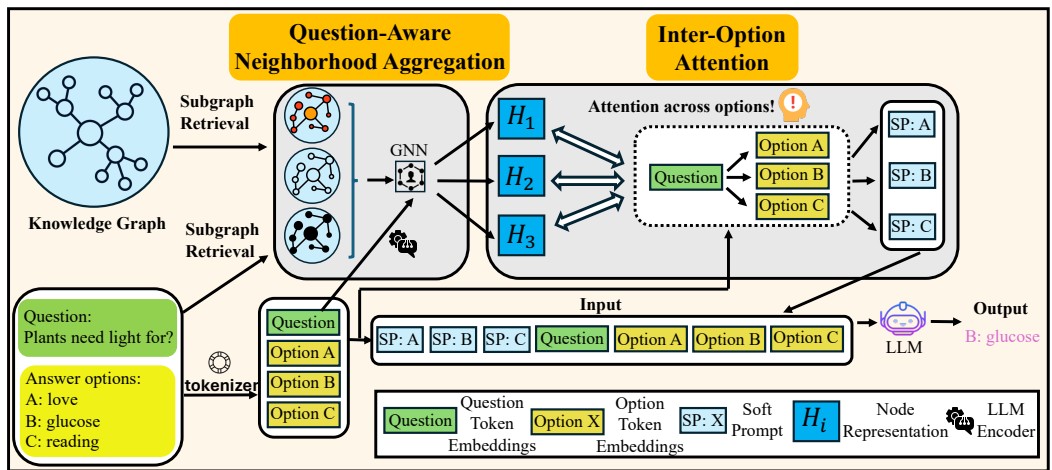

Figure 2: Overview of our proposed framework QAP. The framework consists of: (1) Subgraph Retrieval, where contextualized subgraphs from the KG are extracted based on the question and answer options; (2) Question-Aware Neighborhood Aggregation (QNA), where the KG is processed with neighborhood aggregation influenced by the question context; (3) Inter-Option Attention (IPT), which refines the node representation output by QNA via aligning KG information using token embeddings and attentions across options. Finally, the refined embeddings, i.e., the soft prompts, are used to guide the LLM in predicting the correct answer.

## 3    QAP

In this section, we introduce the details of our proposed framework QAP. As presented in Figure 2, QAP is structured into three phases: *(i)* Subgraph Retrieval, *(ii)* Question-Aware Neighborhood Aggregation (QNA) and *(iii)* Inter-Option Attention (IPT). During the Subgraph Retrieval phase, we extract a contextualized subgraph from the KG, containing the information relevant to the question and each answer option. In the Question-Aware Neighborhood Aggregation phase, we utilize a specialized GNN, where the aggregation process is impacted by the question, allowing the subgraph processing to generate outputs that are more closely aligned with the query. Finally, in the Inter-Option Attention phase, we employ an attention module to capture the relationships between different options and transfer the output of the GNN into a form that is easier for LLM to understand and process. The outputs of IPT serve as soft prompts to guide LLMs. Notably, the entire framework is optimized in an end-to-end manner and no intermediate training objective is needed.

### 3.1    SUBGRAPH RETRIEVAL

To effectively utilize and retrieve the useful information in the KG that is relevant to the given question, we extract the contextualized subgraphs of the questions to reduce the size of the KG used while capturing useful data. Specifically, for an answer option $a_k$, we first establish the set of all entities in $\mathcal{G}$ that appears in the question $q$ and answer option $a_k$, denoted as $\mathcal{E}_q^k$. We then extract the $N$-hop neighbors and the relation connecting them as the contextualized subgraph of $a_k$, denoted as $\mathcal{G}_q^k$ (Yasunaga et al., 2022). This contextualized subgraph contains the necessary information that is potentially helpful for determining whether the option $a_k$ is correct for the given question $q$.

### 3.2    QUESTION-AWARE NEIGHBORHOOD AGGREGATION

After obtaining the contextualized subgraphs during the Subgraph Retrieval phase, we introduce a Question-Aware Neighborhood Aggregation mechanism (QNA) within each subgraph $\mathcal{G}_q^k$, guided by the question embedding given by LLM. The goal is to generate node representations that not only capture the structural properties of the KG but also emphasize the triplet relevant to the question $q$, thus making the final output more compatible with the question.

**Question-Aware Attention Mechanism.** QNA uses a specialized GNN involving question-relevance assessment. In this model, an attention mechanism is employed to incorporate the relevance between knowledge graph entities and the question $q$ to the aggregation weight. To enhance the model's capacity to focus on different parts of the node features, we use a multi-head attention mechanism in the model.

Let $\mathbf{X} \in \mathbb{R}^{N_k \times d_g}$ denote the node feature matrix for the subgraph $\mathcal{G}_q^k$, which is initially the pre-trained entity embeddings. $N_k$ is the number of nodes in $\mathcal{G}_q^k$ and $d_g$ is the feature dimension. The question $q$ is encoded into an embedding $\mathbf{q} \in \mathbb{R}^{d_t}$ by $LM$, which is used to guide the attention mechanism within the Question-Aware Neighborhood Aggregation. $d_t$ is the dimension of the embeddings of $LM$.

In a layer of GNN, specifically, for each neighboring node pair $(i, j)$ within the subgraph $\mathcal{G}_q^k$, we compute attention over $H$ different heads. We denote $\mathbf{Q}_i^h = \mathbf{W}_Q^h \mathbf{X}_i$ and $\mathbf{K}_j^h = \mathbf{W}_K^h \mathbf{X}_j$ are respectively the query and key vectors for nodes $i$ and $j$ in the $h$-th head, with $\mathbf{X}_i$, $\mathbf{X}_j$ being the feature vector of node $i$, $j$ in the current GNN layer. Here, $\mathbf{W}_Q^h$ and $\mathbf{W}_K^h$ are respectively the learnable weight matrices for the query and key transformations in the $h$-th head. For the $h$-th attention head, we have three attention components, $\alpha_{ij,h}^{(1)}$, $\alpha_{ij,h}^{(2)}$, and $\alpha_{ij,h}^{(3)}$, which are computed as follows, where $d_k$ is the dimension of the key vectors for each head:

- **Node-to-Node Attention:**

$$\alpha_{ij,h}^{(1)} = \frac{\mathbf{Q}_i^h \cdot \mathbf{K}_j^h}{\sqrt{d_k}}. \tag{1}$$

- **Question-to-Node Attention:**

$$\alpha_{ij,h}^{(2)} = \frac{\mathbf{Q}_q^h \cdot \mathbf{K}_j^h}{\sqrt{d_k}}, \tag{2}$$

where $\mathbf{Q}_q^h = \mathbf{W'}_Q^h \mathbf{q}$ is the query vector derived from the question embedding $\mathbf{q}$ in the $h$-th head.

- **Node-to-Question Attention:**

$$\alpha_{ij,h}^{(3)} = \frac{\mathbf{Q}_i^h \cdot \mathbf{K}_q^h}{\sqrt{d_k}}, \tag{3}$$

where $\mathbf{K}_q^h = \mathbf{W'}_K^h \mathbf{q}$ is the key vector derived from the question embedding $\mathbf{q}$ in the $h$-th head.

Here $\mathbf{W'}_Q^h$ and $\mathbf{W'}_K^h$ are learnable weights. The attention components $\alpha_{ij,h}^{(1)}$, $\alpha_{ij,h}^{(2)}$, and $\alpha_{ij,h}^{(3)}$ are then weighted summed and applied to $\mathrm{softmax}$:

$$\alpha_{ij,h} = (1 - 2\gamma)\alpha_{ij,h}^{(1)} + \gamma\alpha_{ij,h}^{(2)} + \gamma\alpha_{ij,h}^{(3)}, \tag{4}$$

$$\tilde{\alpha}_{ij,h} = \frac{\exp(\alpha_{ij,h})}{\sum_l \exp(\alpha_{il,h})} \tag{5}$$

where $\gamma \in (0, 0.5)$ is the weight for the impact of the question on aggregations. The attention output for each head is then computed as:

$$\mathbf{Z}_i^h = \sum_{j \in \mathcal{N}(i)} \tilde{\alpha}_{ij,h} \mathbf{V}_j^h, \quad \text{where } \mathbf{V}_j^h = \mathbf{W}_V^h \mathbf{X}_j, \tag{6}$$

Here $\mathbf{W}_V^h$ is the learnable weight for the value transformation in the $h$-th head. Finally, the outputs from all heads are concatenated and linearly transformed to update the node feature of the $i$-th node:

$$\mathbf{X}_i' = \mathbf{W}_O \left[ \mathbf{Z}_i^1 \| \mathbf{Z}_i^2 \| \dots \| \mathbf{Z}_i^H \right] + \mathbf{X}_i, \tag{7}$$

where $\mathbf{W}_O$ is the learnable weight and $\|$ denotes concatenation. $\mathbf{X}_i'$ is used as the input feature in the next GNN layer. These node representations in the final GNN layer, enriched with both structural and question-relevant information, are used in subsequent phases to generate soft prompts for the LLM. The process of using these node presentations to assist the LLM in answering the question will be detailed in the following section.

## 3.3 INTER-OPTION ATTENTION

In this subsection, we describe the Inter-Option Attention (IPT) mechanism. IPT incorporates the relationships among different options to soft prompt generating and transforms the output of QNA into a form that is more interpretable by the LLM. The idea is to align the GNN node representations from each contextualized subgraph with the token embeddings of all options along with the question, which contain text information interpretable by LLM across all options.

**Cross-Option Node-Token Attention.** After processing each subgraph $\mathcal{G}_q^k$ through QNA, we obtain node representations for each node in the subgraph. Let $\mathbf{H}_k \in \mathbb{R}^{N_k \times d_g}$ denote the node representations for the subgraph corresponding to the answer option $a_k$. Additionally, for the question $q$ and its $n$ answer options, we construct $n$ different sequences $\mathbf{T}_1, \mathbf{T}_2, \ldots, \mathbf{T}_n$, where each sequence $\mathbf{T}_r$ is a concatenation of the of token embeddings from question $q$ and the $r$-th answer option $a_r$. We denote $\mathbf{T}_r = \{\mathbf{T}_{r,1}, \mathbf{T}_{r,2}, \ldots, \mathbf{T}_{r,m}\}$, where $m$ is the number of tokens in $\mathbf{T}_r$ and each token embedding $\mathbf{T}_{r,s} \in \mathbb{R}^{d_t}$, with $d_t$ as the token embedding dimension. To ensure compatibility between the node and token embeddings, we first project these embeddings into the same dimensional space using a linear transformation. For the $i$-th node in $\mathbf{H}_k$ and the $s$-th token in $\mathbf{T}_r$:

$$\mathbf{H}'_{k,i} = \mathbf{W}_{P_g}\mathbf{H}_{k,i}, \quad \mathbf{T}'_{r,s} = \mathbf{W}_{P_t}\mathbf{T}_{r,s}, \tag{8}$$

where $\mathbf{W}_{P_g}$ and $\mathbf{W}_{P_t}$ are the projection matrices. Next, we perform an attention operation where each node embedding serves as the query, and the token embeddings $\mathbf{T}'_{r,s}$ serve as both the keys and values. For each answer option $a_k$, we use $n$ separate attention heads. Each head corresponds to one of the $n$ token embedding sequences. Specifically, for the $r$-th head, the attention between the $i$-th node embedding and the $s$-th token embedding is computed as follows:

$$\beta_{is}^{(r)} = \frac{\exp\left(\frac{\mathbf{H}'_{k,i} \cdot \mathbf{T}'_{r,s}}{\sqrt{d_t}}\right)}{\sum_{u=1}^{m} \exp\left(\frac{\mathbf{H}'_{k,i} \cdot \mathbf{T}'_{r,u}}{\sqrt{d_t}}\right)}, \tag{9}$$

where $\beta_{is}^{(r)}$ represents the attention weight between node $i$ in subgraph $\mathcal{G}_q^k$ and token $s$ in the $r$-th text sequence. The resulting attention weights $\beta_{is}^{(r)}$ are then used to compute a weighted sum of the token embeddings for the $r$-th head, yielding a new representation for each node:

$$\tilde{\mathbf{H}}_{k,i}^{(r)} = \sum_{s=1}^{m} \beta_{is}^{(r)}\mathbf{T}'_{r,s}. \tag{10}$$

Finally, the outputs from all $n$ heads are concatenated as the final representation for each node:

$$\hat{\mathbf{H}}_{k,i} = \mathbf{W}_{O_t}\left[\tilde{\mathbf{H}}_{k,i}^{(1)}\|\tilde{\mathbf{H}}_{k,i}^{(2)}\|\ldots\|\tilde{\mathbf{H}}_{k,i}^{(n)}\right]. \tag{11}$$

Here $\mathbf{W}_{O_t}$ is the output weight matrix. This process converts each node embedding into a distribution that not only approximates the token embedding space, but also incorporates information from multiple text sequences corresponding to different answer options. This enables the model to leverage inter-option relationships during the decision-making process.

**Soft Prompt Construction.** Once we have transformed the node representations of each subgraph $\mathcal{G}_q^k$ into the token space, we perform a pooling operation to aggregate embeddings across all nodes in the subgraph. This pooling operation generates a single embedding for each subgraph:

$$\hat{\mathbf{h}}_k = \text{Pooling}\left(\{\hat{\mathbf{H}}_{k,i} \mid i = 1, 2, \ldots, N_k\}\right). \tag{12}$$

Given that there are $n$ answer options, this process results in $n$ pooled embeddings, one for each subgraph. These $n$ embeddings are then prepended to the original token embeddings of the question to form the soft prompts:

$$\mathbf{S}_p = \{\hat{\mathbf{h}}_1, \hat{\mathbf{h}}_2, \ldots, \hat{\mathbf{h}}_n\}. \tag{13}$$

The resulting sequence $\mathbf{S}_p$ serves as the soft prompts appending to the input to the LLM, guiding the LLM to produce an output that is more aligned with the knowledge provided by the KG and tailored to the specific question. The final LLM output is then used to determine the correct answer option. This attention-based transformation effectively bridges the gap between the structured information in the KG and the sequential processing of the LLM and enriches the prompt with inter-option attention, enabling a more contextually aware generation of answers.

## 4 OPTIMIZATION

The optimization of our proposed framework QAP is focused on aligning the LLM's output with the correct answer. Let $\mathbf{y}$ denote the ground truth text associated with the correct answer option. The loss function used to optimize the QAP model is the cross-entropy loss as:

$$\mathcal{L} = -\log P(\mathbf{y}|\mathbf{S}_p + \mathbf{Q} + \mathbf{A}). \tag{14}$$

Here $\mathbf{Q}$, $\mathbf{A}$ represent the token embeddings of the question $q$ and all options $a_1, a_2, \cdots, a_n$. This loss function is used to adjust the parameters of the Question-Aware Neighborhood Aggregation and the Inter-Option Attention module in an end-to-end manner while keeping the LLM parameters frozen. By minimizing the cross-entropy loss, the QAP model learns to produce outputs that are increasingly aligned with the ground truth text, thereby improving its ability to generate soft prompts that can effectively guide the LLM to generate the correct textual output based on the information provided by the knowledge graph and the associated question context.

## 5 EXPERIMENTS

In this section, we introduce the experiments conducted on three datasets of MCQA tasks to demonstrate the effectiveness of the proposed method QAP. We also give an ablation study to evaluate each module of QAP, and a case study to show the performance of QAP. A parameter study on $\gamma$ is shown in Appendix A.3.

### 5.1 DATASETS

We evaluate our model on both general domain and biomedical domain MCQA datasets, utilizing different knowledge graphs for each domain. For the general domain, we use **OBQA** (Open-BookQA) (Mihaylov et al., 2018) and **Riddle** (RiddleSense) (Lin et al., 2021) with Concept-Net (Speer et al., 2017) as the background knowledge graph. For the biomedical domain, we test QAP on **MedQA** (MedQA-USMLE) (Jin et al., 2021) dataset with KG Unified Medical Language System (UMLS) (Bodenreider, 2004). We introduce these datasets in Appendix A.1.

### 5.2 BASELINES

We compare the performance of our proposed method QAP with the following five baselines:

- **LLM (LLM-Only)**: This baseline uses the Large Language Model directly to answer the questions, without any additional prompt enhancements or external knowledge integration.
- **PE (Prompt-Enhanced LLM)**: In this method, we utilize the same LLM but with designed prompts to guide the model's reasoning. These prompts are crafted to better align with the specific requirements of the question.
- **KGEP (KG Evidence Prompting)** (Baek et al., 2023; Liu et al., 2024): This approach incorporates Knowledge Graph triplets into the prompt. A decoder first ranks the similarity between KG triplets and the given question, selecting the highest-scoring triplets as evidence. These triplets are then added to the prompt to aid the LLM in generating a more informed answer.
- **SP (Soft Prompting)** (Lester et al., 2021): We train soft prompts without utilizing any external KG information. These soft prompts assist the LLM in generating answers and the knowledge comes solely from the pretrained language model without external knowledge.
- **GNP (Graph Neural Prompting)** (Tian et al., 2024): GNP uses a graph neural network to encode KG information into the LLM's prompts. GNP consists of a GNN encoder, a cross-modality pooling module, and self-supervised link prediction to better incorporate KG knowledge into the language model as soft prompts.

### 5.3 EXPERIMENTAL SETTINGS

We implement our method with the 3B and 11B parameter versions of the Flan-T5 model (Wei et al., 2022a) as the large language models. The model performance is evaluated using accuracy. More implementation details are shown in Appendix A.2.

Table 1: Comparison between the accuracy(%) and standard deviation(%) over QAP and baselines on the tasks on the three datasets. The best and second-best results are respectively shown in **bold** and underlined.

| Method | Flan-T5 (3B) | | | Flan-T5 (11B) | | |
|---|---|---|---|---|---|---|
| | OBQA | Riddle | MedQA | OBQA | Riddle | MedQA |
| LLM | 73.60±0.10 | 55.29±0.20 | 34.25±0.04 | 80.40±0.10 | 66.08±0.10 | 39.28±0.08 |
| PE | 75.00±0.50 | 55.88±0.49 | 34.49±0.20 | 83.20±0.30 | 65.29±0.20 | 39.67±0.24 |
| KGEP | 73.00±0.60 | 48.43±1.18 | 30.56±0.47 | 78.00±1.00 | 61.37±1.18 | 34.41±0.50 |
| SP | 75.60±0.40 | 54.31±0.59 | 34.33±0.31 | 84.60±0.20 | 65.29±1.57 | 39.28±0.08 |
| GNP | 76.20±0.80 | 57.06±0.98 | 34.01±0.55 | 85.40±0.80 | 67.84±0.78 | 39.91±0.63 |
| QAP | **82.00**±0.50 | **68.82**±0.20 | **38.57**±1.02 | **88.20**±0.60 | **77.06**±0.39 | **44.30**±0.47 |

## 5.4 RESULTS AND ANALYSIS

In this section, we present the performance of QAP, in comparison to various baselines across the three datasets. The overall performance of QAP and the baselines are presented in Table 1. Our method consistently outperforms the baselines on both the general domain (OBQA and Riddle) and the biomedical domain (MedQA).

QAP shows improvements on OBQA, which relies more on direct factual recall from elementary-level science concepts. This may already be well captured by the LLM itself, reducing the potential benefit from external KG integration. However, QAP still demonstrates the effectiveness of incorporating structured knowledge and multi-head attention mechanisms to enhance reasoning in these cases. Specifically, QAP outperforms the best baseline by 7.61% with the 3B LLM and 3.28% with the 11B LLM. For Riddle, which requires complex commonsense reasoning, the inclusion of ConceptNet and our QNA mechanism enabled better extraction and utilization of relevant knowledge to solve riddles. The ability of IPT to model relationships between options also contributed to improved reasoning, as our method can better infer the correct answer by comparing options against one another. Here, the performance gains were more substantial, with improvements of 20.61% for the 3B LLM and 13.59% for the 11B LLM over the baseline. On MedQA, the use of UMLS as a knowledge graph proved critical. Medical questions often require highly specialized knowledge, and by leveraging UMLS through our method, the model could access domain-specific information not present in standard language models. This integration allowed QAP to better interpret the biomedical context of questions, leading to an improvement of 11.83% with the 3B LLM and 11.00% with the 11B LLM.

These results highlight the importance of leveraging question-aware external structured knowledge and modeling inter-option relationships for enhancing the reasoning capabilities of large language models, particularly in complex domains such as biomedical reasoning.

## 5.5 ABLATION STUDY

We perform three ablation studies to evaluate the contribution of key components in our model, shown in Table 2. First, we remove the Question-Aware Neighborhood Aggregation component (QNA) by excluding the question embeddings and using only KG embeddings for aggregation. This results in a significant drop in accuracy, showing that incorporating question-specific information is critical for guiding the GNN to focus on the most relevant knowledge from the KG. Second, we remove the Inter-Option Attention mechanism (IPT). Without this component, the model is less effective at relating KG information to the question context and can not manage the relations between different options, leading to a noticeable performance decrease. Finally, we evaluate the effect of removing the multiple heads of aggregation in GNA, which reduces the model's ability to capture diverse perspectives from the KG. This leads to further declines in performance. Each of these components is found to play a vital role in the overall performance of our model.

Table 2: Experimental results of ablation studies. This table present the accuracy(%) of the studies and standard deviation(%) on the three datasets. The best results are shown in **bold**, respectively. Here "w/o QNA", "w/o IPT" and "w/o MH" respectively represent the removal of QNA, IPT and multiple heads.

| Method | Flan-T5 (3B) | | | Flan-T5 (11B) | | |
|---|---|---|---|---|---|---|
| | OBQA | Riddle | MedQA | OBQA | Riddle | MedQA |
| QAP | **82.00**±0.50 | **68.82**±0.20 | **38.57**±1.02 | **88.20**±0.60 | **77.06**±0.39 | **44.30**±0.47 |
| QAP w/o QNA | 75.60±1.00 | 63.53±0.20 | 34.87±1.18 | 84.40±1.10 | 68.04±0.29 | 42.26±0.31 |
| QAP w/o IPT | 76.60±0.90 | 63.73±0.29 | 35.19±1.26 | 82.40±0.80 | 66.67±0.39 | 42.73±0.55 |
| QAP w/o MH | 76.40±1.20 | 63.92±0.39 | 35.42±0.86 | 85.00±1.40 | 70.39±0.59 | 43.05±0.24 |

## 5.6 CASE STUDY

To further illustrate the effectiveness of QAP, we conduct a case study by selecting examples from both the general domain (OBQA) and the biomedical domain (MedQA) to compare the next-token prediction results between QAP and the baseline that only uses LLM. For each example, we analyze the LLM's predicted logits (i.e., the scores before applying $\mathrm{softmax}$, represent the model's confidence for each token.) for the next token corresponding to each answer option (A, B, C, D).

In these examples, we find that when only the LLM is used, the highest-score token predicted by the model does not correspond to the correct answer. However, when our method is applied, which incorporates knowledge from KG through QNA and IPT, the correct answer token receives the highest predicted score. This demonstrates the effectiveness of QAP in guiding the model toward more accurate predictions. We present these results visually in Figure 3. In the figure, the scores shift more favorably towards the correct answer when our method is used, further validating the benefit of our method.

## 6 RELATED WORK

### 6.1 LARGE LANGUAGE MODELS AND QUESTION ANSWERING

Large Language Models, such as GPT-3 (Brown et al., 2020) and Flan-T5 (Wei et al., 2022a), have shown remarkable performance across various natural language processing tasks (Wei et al., 2022b), including MCQA (Tian et al., 2024). However, LLMs still face limitations in reasoning tasks that require access to factual knowledge beyond their pre-training corpus (Luo et al., 2024). Several approaches have been proposed to augment LLMs with external knowledge sources, such as knowledge graphs, to enhance their factual accuracy and reasoning capabilities (Baek et al., 2023). For example, methods like Retrieval-Augmented Generation (RAG) have introduced mechanisms to retrieve relevant information from external sources, including KGs, and incorporate it into LLM inputs (Xu et al., 2024; Shi et al., 2024; Wang et al., 2024). While effective in some scenarios, these approaches often struggle with noisy retrievals or insufficiently grounded knowledge, limiting their impact on complex reasoning tasks.

### 6.2 KNOWLEDGE GRAPHS FOR ENHANCING QA

Knowledge graphs provide structured representations of entities and their relationships, making them valuable resources for improving the reasoning abilities of LLMs in knowledge-intensive tasks (Zhang et al., 2019; Ma et al., 2024; Jiang et al., 2024). Prior work, such as QA-GNN (Yasunaga et al., 2021), has demonstrated the effectiveness of using graph neural networks (GNNs) to model the relationships within KGs and integrate these relationships into the question-answering process. These approaches allow LLMs to reason over multi-hop knowledge, bridging gaps in factual knowledge that are not readily accessible through text-based models alone (Jiang et al., 2023b;a). However, many existing methods treat KGs as static resources, retrieving specific facts based on direct entity matches without fully leveraging the contextual relevance of knowledge to the question (Sun et al., 2024). This limitation can reduce the potential of KGs to support deeper reasoning tasks, such as commonsense or biomedical question answering.

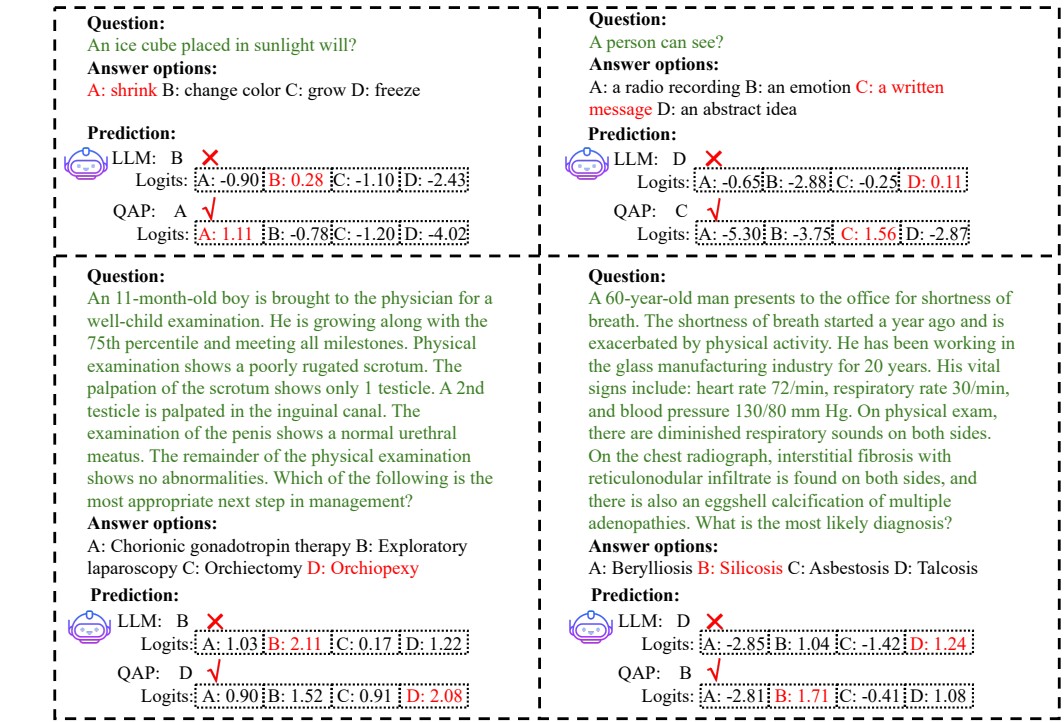

Figure 3: Instances over QAP and LLM-only on both the general and biomedical domains. We list the logits given by LLM and our method QAP. The instances present that QAP provides a more accurate prediction. The correct answer and the option with the highest logit value are shown in red.

## 6.3 GNNs and Soft Prompts for KG Integration

Recent advances in integrating GNNs with LLMs have introduced the use of GNNs to generate soft prompts (Lester et al., 2021; Fang et al., 2023), and guide the LLM's reasoning process by encoding KG information directly into the model's input. For instance, Graph Neural Prompting (GNP) (Tian et al., 2024) incorporates a GNN to learn from KG data and produce neural prompts that enhance the LLM's performance on both commonsense and domain-specific reasoning tasks. GNN-based approaches, however, often rely on static KG structures and fail to incorporate the question directly into the KG aggregation process (Pan et al., 2024; Zhang et al., 2022). This lack of interaction between the question and KG during the aggregation phase can result in suboptimal utilization of the KG, especially for questions requiring nuanced reasoning.

## 7 Conclusion

In this paper, we proposed a novel approach, QAP, to enhance the performance of Multiple Choice Question Answering (MCQA) tasks by integrating Knowledge Graphs with Large Language Models. Our method addresses two key challenges in existing approaches. First, we introduced a Question-Aware Neighborhood Aggregation (QNA) mechanism that incorporates question embeddings into the aggregation process of GNNs, improving the model's ability to assess the relevance of KG information based on the question context. This allows the GNNs to focus on the most relevant knowledge for answering the question to capture useful information. Second, we designed a novel attention mechanism, IPT, enabling inter-option interactions. By utilizing multi-head attention where each head attends to a different answer option, we allowed the model to leverage the relationships between the answer options. This strategy improves the model's ability to eliminate incorrect options and enhance overall performance. Our method was evaluated on three MCQA datasets of two domains, and experimental results demonstrated that QAP outperforms state-of-the-art models, validating its effectiveness and scalability. We believe that integrating structured external knowledge with LLMs through attention and interaction mechanisms will continue to be a promising direction for advancing question answering systems.

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

# A APPENDIX

## A.1 DATASETS

In this subsection, we introduce the data we use to evaluate the proposed method QAP.

- **OBQA (OpenBookQA)** (Mihaylov et al., 2018): A QA dataset focuses on open-book science questions that require reasoning with facts from a set of elementary-level science concepts. This is a 4-way MCQA task containing 5,957 elementary science questions. We use ConceptNet (Speer et al., 2017) as the background knowledge graph to provide external knowledge for reasoning.

- **Riddle (RiddleSense)** (Lin et al., 2021): A dataset designed for commonsense reasoning, where the questions are riddles that require higher-level reasoning skills. It is a 5-way MCQA task testing complex riddle-style commonsense reasoning with 5,715 questions. We use Concept-Net (Speer et al., 2017) as the knowledge graph to support the reasoning process.

- **MedQA (MedQA-USMLE)** (Jin et al., 2021): A QA dataset in the biomedical domain that contains questions from the United States Medical Licensing Examination (USMLE). It is a 4-way MCQA task containing 12,723 United States Medical License Exam questions. For this dataset, we use the Unified Medical Language System (UMLS) (Bodenreider, 2004) as the knowledge graph to provide domain-specific biomedical knowledge.

- **ConceptNet** (Speer et al., 2017): ConceptNet is a general-domain knowledge graph representing general human knowledge in the form of semantic relationships between words and phrases (concepts), containing 799,273 nodes and 2,487,810 edges.

- **UMLS (The Unified Medical Language System)** (Bodenreider, 2004): UMLS is a biomedical knowledge graph developed by the U.S. National Library of Medicine, containing 9,958 nodes and 44,561 edges. It integrates multiple medical terminologies and ontologies into a single structured resource. UMLS is particularly valuable for domain-specific tasks where general language models lack sufficient expertise in biomedical knowledge.

## A.2 IMPLEMENTATION DETAILS

We implement our method using PyTorch, with the 3B and 11B parameter versions of the Flan-T5 model (Wei et al., 2022a) as the large language models. For the general domain datasets (OBQA and Riddle), we use ConceptNet as the knowledge graph, and for the biomedical dataset (MedQA), we use UMLS. Contextualized subgraphs are extracted from these KGs including the two-hop neighbors of entities appearing in the question and options to assist in answering questions.

The GNN model in QNA consists of 3 layers with $\gamma = \frac{1}{3}$, followed by the attention module ITP. Soft prompts, with a size of 2048/4096 on different LLMs, are trained end-to-end to enhance LLM performance. We use the AdamW optimizer (Loshchilov & Hutter, 2018) and a learning rate of $5 \times 10^{-6}$ for both the 3B and 11B models. Model performance is evaluated using accuracy.

We provide the source code and the datasets at `https://anonymous.4open.science/r/QAP-13AC`.

## A.3 PARAMETER STUDY

To analyze the impact of the weight distribution among the components in our Question-Aware Neighborhood Aggregation module (QNA), we conducted a parameter study on Flan-T5 (11B) by varying the weight distribution among the three key components in the aggregation process: Node-to-Node, Question-to-Node, and Node-to-Question. In this study, we adjusted the weight distribution using the parameter $\gamma$. The weight distribution is $(1 - 2\gamma)$ for Node-to-Node, and $\gamma$ for both Question-to-Node and Node-to-Question, which are question-related interactions.

We evaluated the effect of $\gamma$ on both OBQA and MedQA datasets, representing the general and biomedical domains, respectively. The results, shown in Figure 4, indicate that the optimal value of $\gamma$ differs slightly between the two domains. For OBQA, the model achieves its best performance with $\gamma$ around 0.2, whereas for MedQA, the optimal $\gamma$ is closer to 0.4.

In both cases, the results suggest that a balance between Node-to-Node and question-related interactions is crucial for optimal performance. When $\gamma$ is set too low, the model over-relies on Node-to-Node interactions, failing to fully capture the relevance of the question to the knowledge graph, which is particularly important for complex reasoning tasks. Conversely, when $\gamma$ is set too high, giving excessive weight to question-related interactions, the model loses the structural information inherent in the knowledge graph, which is essential for retaining factual consistency.

For general-domain datasets like OBQA, giving slightly more emphasis to the Node-to-Node interactions helps retain important structural information from the knowledge graph, which aligns with the nature of the questions that often require factual recall. These questions tend to focus on basic scientific concepts, and retaining the KG's structural integrity allows the model to effectively leverage the factual relationships between entities. In contrast, for biomedical-domain datasets like MedQA, increasing the weight on question-related interactions enhances the model's ability to leverage the question context for more complex, domain-specific reasoning. Biomedical questions often involve intricate relationships and specific terminology, where aligning the KG information with the question context becomes crucial for accurate reasoning. As a result, in MedQA, the best performance is observed when $\gamma$ is set to about 0.4, giving more weight to the question-related components of the GNN aggregation, and allowing the model to focus more on question-specific entities and their relationships.

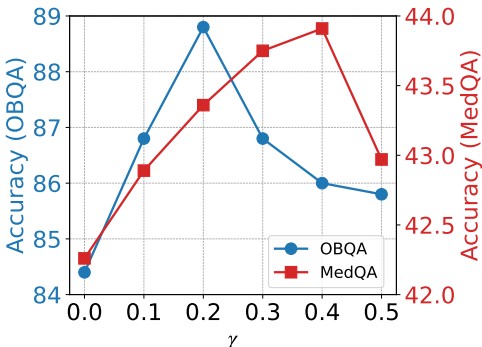

Figure 4: Effect of varying $\gamma$ on model performance for OBQA (general domain) and MedQA (biomedical domain). Results suggest that a balanced combination of structural and question-context information is crucial for optimal reasoning across different domains.

### A.4 LIMITATIONS

While our proposed method demonstrates significant improvements in Multiple-Choice Question-Answering (MCQA) tasks by integrating knowledge graphs and leveraging Question-Aware strategies, several limitations still remain. First, our approach relies heavily on the quality and completeness of the external knowledge graph. In domains where the KG is sparse or lacks coverage, such as less-studied areas or questions involving uncommon entities, the model's performance may degrade. Moreover, our method does not explicitly handle cases where external knowledge is ambiguous or conflicts with the question context, potentially leading to confusion in the model's final predictions. Second, the computational complexity of our design can lead to increased inference time, making it less suitable for real-time applications or deployment in resource-limited environments.

### A.5 ETHICS STATEMENT

Our work focuses on improving the performance of large language models for Multiple-Choice Question Answering by integrating structured knowledge from knowledge graphs. While our method enhances the factual accuracy and reasoning capabilities of LLMs, we acknowledge several ethical considerations. First, the use of external knowledge sources such as knowledge graphs introduces potential biases inherent in the data. Knowledge graphs may reflect the biases of their creators, including historical, cultural, and societal biases, which could inadvertently affect the fairness and neutrality of the model's predictions. Second, in sensitive domains such as healthcare (e.g., MedQA), the reliance on imperfect knowledge graphs may lead to incorrect or harmful predictions, especially in situations where the information in the KG is outdated or incomplete. This underscores the importance of validation and continuous updating of the external knowledge sources used by the model. We are committed to promoting fairness and effectiveness in AI, and we encourage the responsible use of our method, particularly in high-stakes applications.

