# OpenReview forum: "Question-Aware Knowledge Graph Prompting for Large Language Models"
_ICLR.cc/2025/Conference — ICLR 2025 Conference Withdrawn Submission_

### Official Review · Reviewer_KeZ3 · 2024-10-16

**Soundness:** 3
**Presentation:** 2
**Contribution:** 2
**Rating:** 3
**Confidence:** 5

**Summary:**

They addressed the problem of Knowledge Graph (KG)-based multiple-choice question answering (MCQA). In this approach, a soft prompt manner utilizing Graph Neural Networks (GNNs) was employed. They criticized existing GNN-based methods for not adequately considering the context of the question and proposed new modules to address this limitation. The proposed modules are Question-Aware Neighborhood Aggregation (QNA) and Inter-Option Attention (IPT). QNA performs attention operations among the nodes of the subgraph, the question, and answer candidates, while IPT executes attention operations between the question's linguistic representation and the graph representation. Experiments were conducted using Flan-T5, and significant performance improvements in MCQA were achieved by training only the prompt modules without further training the LLM.

**Strengths:**

The new modules, IPT and QNA, contributed to enhancing MCQA performance. By proposing a prompt method specialized for multiple-choice problems, this work has facilitated constructive discussion in the study of this task. The fact that a relatively simple approach led to significant performance improvements is noteworthy.

**Weaknesses:**

1. QA-GNN [1] has already discussed the issue of graph embedding that incorporates question context. However, this study does not provide a comparison with that work.
2. GNP [2] also proposed a method for combining question context and graph embedding using cross-modality pooling, which is quite similar to the QNA proposed by the authors. Nevertheless, only a simple performance comparison is presented, and there is no conceptual comparison.
3. Although an ablation study is included, a conceptual explanation is needed to clarify why the proposed method performs better than GNP [2].
4. Recently, large language models have predominantly been released as decoder-only architectures. The authors conducted experiments using Flan-T5, which is an encoder-decoder architecture, but it is necessary to experimentally verify whether the proposed method can also function effectively in decoder-only architectures.

[1]Yasunaga, M., Ren, H., Bosselut, A., Liang, P., & Leskovec, J. (2021). QA-GNN: Reasoning with language models and knowledge graphs for question answering. arXiv preprint arXiv:2104.06378.

[2] Tian, Y., Song, H., Wang, Z., Wang, H., Hu, Z., Wang, F., ... & Xu, P. (2024, March). Graph neural prompting with large language models. In Proceedings of the AAAI Conference on Artificial Intelligence (Vol. 38, No. 17, pp. 19080-19088).

**Questions:**

1. What are the conceptual differences between GNP and your approach, and what do you consider to be the key factors that lead to higher performance compared to GNP?
2. Could you explain the conceptual differences between the Query Node discussed in QA-GNN and QAP in detail?
3. While the proposed method is specialized for MCQA, is it possible to extend and apply it to general QA?
4. Is there a specific reason why the Decoder-Only architecture was not used in the experiments?

---

### Official Review · Reviewer_X7Ab · 2024-10-29

**Soundness:** 2
**Presentation:** 3
**Contribution:** 2
**Rating:** 3
**Confidence:** 5

**Summary:**

The paper proposes Question-Aware Knowledge Graph Prompting (QAP), which incorporates the question context and interactions between options into soft GNN prompting. QAP has been applied on multiple MCQA tasks, claiming to surpass current SOTA models.

**Strengths:**

The proposed method, QAP, mainly consists of two modules: Question-Aware Neighborhood Aggregation (QNA) and Inter-Option Attention mechanism (IPT) while keeping the LLM backbone frozen. QAP outperforms other prompting methods, and ablation studies have shown that all design choices in QAP are reasonable.

**Weaknesses:**

- **More LLM backbones are needed**: The proposed method only experiments with Flan-T5 models in 3B and 11B, which could barely support the paper's claim on QAP. Please try more LLMs, such as LLaMA.

- **Incompleteness of case study**: Based on your implementation, it should not be difficult to apply it to all test datasets and visualize the results. Merely giving four examples is not solid enough for the case study.

- **More details in experimental settings and fair comparison between ablations**: Authors are training models by forcing them to generate the ground texts of correct options. In evaluation/inference, it seems that next-token logits are used to get the predicted option, which overlooks the LLM reasoning processes [1] and makes one of the ablation settings, i.e., Prompt-Enhanced LLM, perform lower.


[1] Wang X, Ma B, Hu C, et al. " My Answer is C": First-Token Probabilities Do Not Match Text Answers in Instruction-Tuned Language Models[J]. arXiv preprint arXiv:2402.14499, 2024.

**Questions:**

- **Typos**: Line 73: double ''the'', Line 376: ''GNA'' should be ''QNA'',

---

### Official Review · Reviewer_AmLp · 2024-11-01

**Soundness:** 1
**Presentation:** 2
**Contribution:** 1
**Rating:** 3
**Confidence:** 4

**Summary:**

his work aims at integrating knowledge graphs into large language models for multi-choice question answering. The authors propose a method called QAP that provides the question-relevance assessment in Question-Aware Neighborhood Aggregation model (QNA) and introduces an Inter-Option Attention module (IPT) to generate KG-based soft prompt. Through the experiments on benchmarks of MCQA, the overall performance of QAP is positive.

**Strengths:**

This paper proposes deep interactions between GNNs and LLMs. Through the experiments on various datasets, the overall performance of QAP is positive and the analyses on each component of QAP (i.e., QNA, IPT, and MH) is extensive.

**Weaknesses:**

1.	There has been extensive research on deep interaction between GNNs and LMs (e.g., the Modality Interaction Layer of GreaseLM (Zhang et al, 2022) and the Cross-Modal Relative Position Bias of QAT (Park et al, 2023)). An extensive comparison should be performed and discussed.
2.	Through the experimental results on OBQA and MedQA with T5-3B, there is no significant improvements between QAP and traditional MCQA model (e.g., QA-GNN, GreaseLM) using small LMs (i.e., AristoRoBERTA and SapBERT). It’s hard to determine whether the superiority of QAP over GNN+LM is solely due to the improved performance of LLMs.
3.	From the characteristics of multi-choice questions, it seems more appropriate to use ToG for multi-step reasoning on KGs since we have already located the topic entity and answer entities. Moreover, the retrieval cost of directly retrieving the N-hop subgraph of the topic entity is much higher than alternately performing beam search-based pruning and sing-hop KG retrieval. ToG also eliminates dependence on pre-trained entity embedding, allowing it to generalize to other KGs. The performance of FlanT5-11B + ToG, and more details about the training and inference cost should be discussed.

**Questions:**

1.	There has been extensive research on deep interaction between GNNs and LMs (e.g., the Modality Interaction Layer of GreaseLM (Zhang et al, 2022) and the Cross-Modal Relative Position Bias of QAT (Park et al, 2023)). An extensive comparison should be performed and discussed.
2.	Through the experimental results on OBQA and MedQA with T5-3B, there is no significant improvements between QAP and traditional MCQA model (e.g., QA-GNN, GreaseLM) using small LMs (i.e., AristoRoBERTA and SapBERT). It’s hard to determine whether the superiority of QAP over GNN+LM is solely due to the improved performance of LLMs.
3.	From the characteristics of multi-choice questions, it seems more appropriate to use ToG for multi-step reasoning on KGs since we have already located the topic entity and answer entities. Moreover, the retrieval cost of directly retrieving the N-hop subgraph of the topic entity is much higher than alternately performing beam search-based pruning and sing-hop KG retrieval. ToG also eliminates dependence on pre-trained entity embedding, allowing it to generalize to other KGs. The performance of FlanT5-11B + ToG, and more details about the training and inference cost should be discussed.

---

### Note · Authors · 2024-11-22

I have read and agree with the venue's withdrawal policy on behalf of myself and my co-authors.